# Sustainable Cultivation and Functional Bioactive Compounds of *Auricularia* Mushrooms: Advances, Challenges, and Future Prospects

**DOI:** 10.3390/biology14111555

**Published:** 2025-11-06

**Authors:** Miao Liu, Wenxin Jiang, Kai Huang, Ling Li, Qingzhong Meng, Xiaoxuan You, Kunlun Pu, Meijing Cheng, Zhenpeng Gao, Jianzhao Qi, Minglei Li

**Affiliations:** 1College of Soil and Water Conservation Science and Engineering, Northwest A&F University, Yangling, Xianyang 712100, China; 2Center of Edible Fungi, Northwest A&F University, Yangling, Xianyang 712100, China; 3Research Center for Eco-Environmental Sciences, University of Chinese Academy of Sciences, Yangling, Xianyang 712100, China; 4Shaanxi Key Laboratory of Natural Products & Chemical Biology, College of Chemistry & Pharmacy, Northwest A&F University, Yangling, Xianyang 712100, China

**Keywords:** *Auricularia* mushrooms, bioactive compounds, cultivation techniques, industrial applications

## Abstract

**Simple Summary:**

*Auricularia* mushrooms are a popular edible fungus with a long history of cultivation and are valued for their potential health benefits. However, there is still a lack of comprehensive understanding regarding their full potential. This review aims to bridge this knowledge gap by systematically compiling and analyzing existing research on these mushrooms. We explore their biological characteristics, cultivation methods, and the various beneficial compounds they contain, such as polysaccharides and melanin, which are known to help with blood sugar control, immune support, and antioxidant activity. The study also highlights their expanding applications beyond food, including uses in drug delivery systems and eco-friendly materials. We identify key challenges, particularly the need for more sustainable cultivation practices. By summarizing the current state of knowledge and future prospects, this work provides a valuable resource for researchers, farmers, and industries, ultimately promoting the broader and more sustainable use of *Auricularia* mushrooms for improving human health and advancing biotechnology.

**Abstract:**

The genus *Auricularia*, and specifically the species *A. auricula*, is a globally significant edible fungus with a long history of cultivation and notable nutritional and medicinal properties. This review systematically examines the taxonomic classification, morphological and physiological characteristics, and bioactive components (such as polysaccharides, melanin, proteins and polyphenols) of *A. auricula*, as well as their pharmacological effects and industrial applications. Recent molecular biological advances have clarified taxonomic uncertainties, including the reclassification of ‘heimuer’ as *A. cornea*, and emphasized the species’ genetic diversity. *A. auricula* thrives in temperate and subtropical regions, with cultivation techniques evolving from traditional wood log inoculation to modern substrate-based methods. However, sustainability challenges persist, including reliance on virgin wood substrates and the need for improved spent substrate management. The fungus exhibits remarkable nutritional properties, with polysaccharides (up to 66.1% of dry weight) demonstrating hypoglycemic, antitumor, immunomodulatory, and antioxidant activities. Melanin and proteins further contribute to hepatoprotection, antimicrobial effects, and metabolic regulation. Industrial applications of *Auricularia* species extend beyond food into pharmaceuticals and functional materials. Polysaccharides are explored as drug carriers, while melanin shows promise in antioxidant and antibacterial formulations. Despite these advances, gaps remain in understanding the mechanistic basis of bioactive compound functions and optimizing cultivation for sustainable production. Future research should integrate multi-omics approaches to elucidate genetic regulation, enhance substrate formulations, and develop value-added products. This review underscores the potential of *Auricularia* species as a functional food and biotechnological resource, advocating for interdisciplinary efforts to address current challenges and unlock its full industrial potential.

## 1. Introduction

Socioeconomic development and increased public health awareness have led to a greater emphasis on optimizing dietary structures and developing functional foods. Within food science and biomedicine, one area of significant interest is the study of edible mushrooms, largely due to their unique nutritional and medicinal properties. For instance, the bioactive molecules present in *Lentinula edodes* and *Pleurotus ostreatus*, including polysaccharides, glucans, ergosterol, amino acids (arginine and glutamine), and proteoglycans, have been shown to possess hypolipidemic, antithrombotic, and anticancer properties [1,2]. Derivatives of a novel type, synthesized from *Hericium erinaceus*, have been demonstrated to possess neuroprotective and immunomodulatory properties. These derivatives include cyanane diterpenes [3]. These mushrooms are increasingly integrated into dietary and pharmaceutical applications, underscoring their potential to bridge nutrition and healthcare.

Despite being the third most cultivated edible fungus in the world—accounting for approximately 17% of total mushroom production—systematic research into the food and medicinal value of *Auricularia auricula* remains relatively limited [4]. Recent advancements in molecular biology have enabled the reclassification of various *Auricularia* species, uncovering widespread synonymy issues in traditional nomenclature For instance, “heimuer” is now recognized as *Auricularia cornea*, rather than *Auricularia polytricha* [5]. Moreover, bioactive components in *A. auricula*, such as polysaccharides and melanin, show significant potential in antioxidant, antitumor, and metabolic regulation activities. Their applications extend beyond traditional food products to include pharmaceutical carriers and functional materials.

This review aims to systematically consolidate and critically evaluate the current body of knowledge on *Auricularia* species, with a particular focus on *A. auricula*. We seek to bridge existing knowledge gaps by addressing four interconnected themes: (1) recent taxonomic reclassifications and genomic insights that redefine species boundaries and phylogenetic relationships; (2) the evolution and sustainability challenges of cultivation technologies, emphasizing substrate optimization and waste management; (3) the pharmacological mechanisms of key bioactive compounds, particularly their roles in metabolic and immune regulation; and (4) the expanding frontier of industrial applications, from functional foods and pharmaceutical carriers to eco-friendly materials. By synthesizing advances across these domains, this work provides a comprehensive framework to guide future research, promote sustainable utilization, and unlock the full potential of *Auricularia* species in enhancing food security, advancing biotechnology, and improving human health.

## 2. Morphological and Physiological Characteristics

Fresh *Auricularia auricula* fruiting bodies exhibit a gelatinous to cartilaginous texture, displaying pronounced hygroscopic properties—swelling upon hydration and rigidifying upon dehydration. Morphological variations range from cup-shaped to ear-like or irregular forms, with growth patterns that may be solitary or clustered. The dorsal (infertile) surface is convex and tomentose, whereas the ventral (fertile) surface is concave, smooth, or wrinkled, bearing basidiospores. Spores are reniform or cylindrical, hyaline, and typically contain one to several oil droplets [6]. Mycelium consists of tubular hyphae that branch and intertwine, forming a white, tree-like network that gradually turns brown with age [7]. Optimal mycelial growth occurs between 25 and 30 °C and pH 5.0–7.0, though significant interspecific variation exists in temperature, humidity, and pH tolerance. The carbon-to-nitrogen (C/N) ratio governs different developmental stages: mycelium stage (C/N ~20:1), primordia formation stage (C/N 30–40:1), and fruiting body emergence stage (C/N ~30:1). Reference [8] reported an optimal temperature range of 22–24 °C for efficient fruiting body emergence, characterized by rapid and orderly development. Transmission electron microscopy revealed that mycelial senescence involves mitochondrial swelling, nucleolar disappearance, and cell wall degradation, culminating in programmed cell death [9]. *A. auricula* is a bipolar heterothallic fungus, governed by a single mating type locus with two alleles. Mature fruiting bodies produce basidiospores of compatible mating types. Upongermination, these spores form uninucleate mycelia or conidia. Compatible uninucleate mycelia fuse to form dikaryotic mycelia, which proliferate via clamp connections and ultimately form basidia. Mature basidia produce numerous basidiospores, completing the life cycle (Figure 1) [10,11].

## 3. Artificial Cultivation

China’s cultivation of *A. auricula* dates back 1400 years. The evolution of artificial cultivation can be categorized into four stages: spore natural inoculation, spore liquid spraying, pure strain inoculation on wood logs, and cultivation on substitute substrates (Figure 2). Historical records, such as Su Jing’s *Xin Xiu Ben Cao* (*Newly Revised Materia Medica*) from the Tang Dynasty, describe *A. auricula* growth on mulberry, sophora, paper mulberry, elm, and willow trees, noting the edibility of tender varieties and medicinal uses (e.g., sophora-grown for hemorrhoids). A described cultivation method involved spreading congee on wood and covering it with straw to induce growth, suggesting that pioneers provided nutrients via irrigation for natural spore inoculation. Wang Zhen’s *Nongshu* from the Song/Yuan period recorded a method involving chopping fruiting bodies into grooves, covering with artemisia leaves and soil, and periodic irrigation with rice rinsing water, resulting in fruiting body emergence under warm, humid conditions post-rainfall. The utilization of fungal nutrients and spore mixtures for artificial inoculation represents a more recent development. The invention of the microscope enabled the discovery of microorganisms, paving the way for direct inoculation of pure fungal cultures onto wood logs (e.g., linden). Significant advancements and proliferation occurred with the maturation of substitute substrate cultivation technology [12]. Large-scale cultivation of *A. auricula* is primarily concentrated in South Korea, Japan, and Southeast Asia, while Western nations focus on pharmacological research. European nations and the United States primarily focus on pharmacological research, relying predominantly on imports for consumption. Currently, *A. cornea* and *A. heimuer* are the predominant species cultivated globally. Market germplasm resources originate mainly from wild domestication, with a limited proportion derived from artificial hybridization [13]. *A. cornea* is extensively cultivated in China using both wood log inoculation and substitute substrate methods. Substitute cultivation has gained popularity due to its cost-effectiveness and higher productivity compared to traditional wood log cultivation [14].

At present, the cultivation of *A. cornea* in China predominantly employs alternative production techniques, including open-air ground placement, small arch tunnels, and shed-based hanging bag systems. These methodologies exhibit distinct management practices that can subtly influence the agronomic quality of the resulting fruiting bodies. Generally, cultivation within hanging bag systems is found to yield superior results compared to traditional open-air methods [15,16,17]. The primary substrate utilized for *A. cornea* cultivation is hardwood sawdust; however, the extensive reliance on virgin wood raises significant environmental concerns, particularly related to deforestation. Therefore, it is imperative to enhance substrate formulations through the inclusion of agroforestry byproducts such as bagasse and cottonseed husks to mitigate these issues [18,19]. Research indicates that a substrate composed of 32% corn stover, 18% corn kernels, and 32% wood shavings effectively supports rapid mycelial colonization of *A. cornea* [18]. Additionally, *A. polytricha* can be cultivated using waste wood shavings from other mushroom species (e.g., *P. eryngii*, *P. cystidiosus*) [20]. This approach proves advantageous, considering that mushrooms typically degrade only 40–80% of the substrate components, thus allowing for the reuse of residual spent substrate [21]. It is noteworthy that lignocellulose is not completely decomposed during cultivation; the resultant spent mushroom substrate (SMS) consists primarily of residual nutrients, fungal mycelium, partially decomposed lignocellulosic biomass, and a variety of organic and inorganic compounds [22]. The direct application of uncomposted SMS to soil presents considerable risks, as it may introduce pathogenic microorganisms that could disrupt native soil ecosystems. Composting SMS prior to application significantly enhances soil physicochemical properties and microbial communities, thereby reducing environmental impacts and improving resource utilization. The composting process has been shown to increase the content of total humic carbon, humic acid carbon, available phosphorus (P), available potassium (K), and nitrate nitrogen (NO_3_-N). Furthermore, the shifts in microbial community composition before and after composting promote organic matter decomposition and optimize nutrient availability for plant uptake [23]. Moreover, *A. heimuer* exhibits the ability to utilize a wide array of nitrogen sources, including nitrates, urea, proteins, ammonium, ammonia, and amino acids. Its growth is contingent upon relatively high levels of nitrogen, in conjunction with essential inorganic elements such as phosphorus (P), sulfur (S), potassium (K), calcium (Ca), and magnesium (Mg). Consequently, practical cultivation often necessitates the formulation of nutrient matrices that incorporate supplements such as bran, corn meal, corn kernels, calcium carbonate, and calcium phosphate to ensure the availability of micronutrients [24]. Extensive research is directed toward modifying cultivation techniques to enhance the micronutrient content of the fruiting bodies (e.g., selenium (Se), copper (Cu), chromium (Cr), iron (Fe), magnesium (Mg), manganese (Mn), nickel (Ni), zinc (Zn)), thereby addressing dietary deficiencies [25,26,27]. Notably, selenium enrichment has garnered particular attention; for instance, ref. [28] succeeded in isolating Se-enriched strains from selenium-rich soil for fertilizer preparation, and the application of these strains to *A. heimuer* cultivation significantly increased the selenium content of the fruiting bodies.

The production systems of *A. auricula*, despite a cultivation history spanning 1400 years, are presently confronted with three critical sustainability challenges. First, there exists a significant reliance on virgin wood, which constitutes over 80% of the substrate composition. This dependence is fundamentally at odds with the principles of the circular economy, while potential alternatives such as underutilized agricultural residues, including bagasse, remain largely confined to experimental frameworks. Second, the practice of directly applying uncomposted spent substrate to land raises concerns regarding the proliferation of soil pathogens. This issue persists despite the fact that advanced biorefinery techniques, including enzymatic extraction and biochar conversion, have yet to be thoroughly explored within this context. Third, while the enrichment of *A. auricula* with selenium presents a promising avenue, there is a notable deficiency in the mechanistic understanding of trace element bioaccumulation and its associated trade-offs concerning yield and stress tolerance. Future innovations in this field must prioritize the integration of synthetic microbial consortia aimed at upgrading lignocellulosic materials, as well as the application of AI-optimized substrate formulations and gene-edited strains. These advancements could foster simultaneous resource conservation and enhancement of nutraceutical properties. Additionally, while the recycling of spent mushroom substrate (SMS) shows potential, its economic feasibility for large-scale agricultural operations remains to be substantiated. Moreover, conflicting reports on heavy metal accumulation in soils amended with SMS compost highlight the necessity for long-term field trials to ensure safety and efficacy.

## 4. Taxonomy and Global Diversity

The fungus *Auricularia* Bull. ex Juss. is classified within the Basidiomycota, *Agaricomycetes*, *Auriculariales*, *Auriculariaceae* [12]. Initially described as Tremella auricula by Linnaeus, it underwent multiple taxonomic revisions before being reclassified as *Auricularia auricula-judae* by Quélet [13]. This species exhibits a global distribution; however, discrepancies exist among fungal databases regarding species counts. For instance, Index Fungorum lists 173 species within the genus *Auricularia*, while MycoBank recognizes only 50, with many entries classified as synonyms, invalid names, or infraspecific variants—currently, approximately 75 valid names are acknowledged [14]. Despite this, the precise number of species in the genus remains uncertain due to limited systematic research on *A. auricula* [15]. In China, the genus comprises 16 species and one variety. The widely cultivated “heimuer” was historically misidentified as *A. polytricha*, but molecular sequencing has since clarified its classification as *A. cornea* [5]. Conversely, the Jamaican variant of *A. polytricha* has been determined to be synonymous with *A. nigricans*. Notably, the principal Chinese cultivar, *A. heimuer*, was originally believed to be equivalent to the European *A. auricula-judae*; however, phylogenetic studies have ultimately confirmed its status as a distinct species [5]. The advent of molecular biology has significantly advanced fungal taxonomy, facilitating the widespread application of molecular marker techniques, including randomly amplified polymorphic DNA (RAPD) and nucleic acid sequence analysis [16,17,18,19]. The analysis of nucleic acid sequences, particularly within conserved regions, has proven especially effective. The ribosomal DNA (rDNA) of fungi contains conserved regions exhibiting varying rates of evolution, rendering them useful as taxonomic markers. The internal transcribed spacer (ITS) region, for example, displays considerable sequence similarity within species or genera, while maintaining sufficient variation between them for effective species differentiation [29,30]. Analyses that integrate ITS and nuclear large subunit (nLSU) sequences have established that *A. auricula* forms a monophyletic group, thereby resolving previous debates surrounding its purported polyphyly [5]. Nevertheless, the limited resolution of ITS and nLSU in distinguishing closely related species within the genus has necessitated the exploration of more variable gene regions, such as *RPB1*, *RPB2*, and *tef-1*, for taxonomic classification [31,32,33,34,35]. Research conducted on common Central American species, utilizing both ITS and *RPB2*, has revealed that *A. auricula-judae* corresponds to the American fungus *A. auricula* Pamasto & I. Pamasto, while specimens previously identified as the wrinkled fungus *A. delicata* were actually *A. scissa* [36]. As highlighted by recent studies, multigene fragment analysis is critical in addressing the phylogenetic classification challenges within *A. auricula*, particularly in identifying appropriate genetic fragments to resolve interspecific variation [37].

Current research on *A. auricula* morphology and physiology remains predominantly descriptive, with critical mechanistic gaps in environmental adaptation, mating system regulation, and senescence processes. Future studies must integrate multi-omics approaches to decipher strain-specific adaptive thresholds, molecular drivers of life cycle transitions, and ultrastructural biomarkers. This will unlock actionable targets for resilient cultivation and biotechnological exploitation of this genus.

## 5. Genomic and Genetic Research

Genomics represents a comprehensive approach to the study of an organism’s genome, encompassing its structure, evolution, and function [38]. Various edible medicinal fungi, including *S. commune* [39], *C. cinerea* [40], *L. edodes* [41], and Ganoderma species [42], have had their genomes and transcriptomes sequenced. Notably, *A. subglabra* was the first species within the genus *Auricularia* to have its genome sequenced [43], with subsequent genomic data for *A. heimuer* and *A. cornea* published more recently [38,44,45,46,47]. Phylogenetic analysis of genomic data indicates that *A. cornea* is more closely related to *A. subglabra* than to *A. heimuer* [45]. The mitochondrial genome of *A. delicata* exhibits significantsynteny with those of *Auricularia polytricha* and *Auricularia auricula-judae*, although discrepancies are present. Comparative analyses of 14 core protein-coding genes encoded in the mitochondrial genome of *A. delicata* have revealed a greater genetic similarity to *A. heimuer* [48]. Furthermore, a study conducted on two endemic *Auricularia* strains from the Qinling Mountains estimated their divergence time at approximately 4.575 million years ago (MYAs), with an even earlier divergence from *A. subglabra* around 33.537 MYAs [49]. Transcriptomics plays a crucial role in elucidating genome structure and function, shedding light on transcriptional regulation, and identifying genetic networks that govern agronomic traits [38]. Applications of transcriptomics include profiling gene expression under varying spatio-temporal conditions, gene discovery, and the identification of molecular markers. Transcriptomic analyses have generated data from various developmental stages, enabling the exploration of genes involved in substrate utilization and primordia formation. Furthermore, regulatory models of mushroom development have been constructed based on phenotypic variations associated with mutations in transcription factors [39]. For instance, a study sequenced the transcriptome of *A. polytricha* using Illumina technology, identifying 9095 potentially unique expressed sequence tags (ESTs), which were then compared to known fungal transcription factor gene families, allowing for an analysis of stage-dependent gene involvement in different pathways [50]. *Auricularia heimuer* possesses genes encoding for seven laccases and various glycoside hydrolases (GH5, GH7, GH12, GH45), which may underlie its remarkable capacity for lignin and cellulose degradation [51]. Overall, genomic approaches significantly enhance our understanding of the phylogeny of *A. auricula* and illuminate the genetic basis of trait correlations.

Current genomic studies of *A. auricula* primarily focus on structural sequencing and phylogenetic comparisons, yet functional annotation of key genes (e.g., laccases, GH families) remains superficial. Critical gaps persist in understanding (1) the regulatory networks linking transcription factors to agronomic traits, (2) the evolutionary drivers of mitochondrial genome synteny breaks, and (3) the functional significance of species-specific divergence timelines (>30 MYAs). Future efforts must integrate multi-omics data to decode genotype-phenotype relationships, particularly lignin/cellulose degradation mechanisms, enabling targeted strain engineering for bioremediation and biomass conversion.

## 6. Geographic Distribution

Wild *Auricularia* species exhibit a broad distribution across temperate and subtropical forest ecosystems, predominantly found in regions of Asia, Europe, and North America. In China, natural populations are particularly concentrated in areas such as the Greater and Lesser Xing’an Mountains, the Qinba Mountain Range, and the Funiu Mountain Range [5,52]. The geographic distribution and morphological characteristics of *Auricularia auricula* demonstrate notable correlations with climatic and geographic barriers. Populations of the same species exhibit distinct morphological traits in different regions, suggesting their limited geographic ranges. For instance, the Appalachian Mountains may serve as a barrier to the distribution of *A. fuscosuccinea* [53]. Typically, *A. auricula* predominantly colonizes angiosperm wood, although occasional instances of growth on gymnosperms have been observed. Furthermore, the type of vegetation significantly influences the growth patterns of various *Auricularia* species [54]. Geographic distribution of major *Auricularia* species complexes is summarized in Table 1 and Figure 3**.**

## 7. Nutritional Composition and Biological Activity

### 7.1. Main Components

Dried *Auricularia auricula* is predominantly composed of carbohydrates (79.9–93.2%), with additional components including crude protein (6.5–13%), crude fiber (3.5–12.5%), and a minimal fat content (0.48–4.5%) [26,55]. The primary constituents are polysaccharides, which can comprise up to 66.1% of the total composition, including mannan, glucan, pectin, chitin, and cellulose. Notably, *Auricularia auricula* contains 18 amino acids, encompassing all eight essential amino acids, as well as three conditionally essential amino acids—arginine, cysteine, and tyrosine—each of which plays a critical role in child development. Among these, glutamic acid stands out as the most prevalent amino acid, whereas sulfur-containing amino acids such as cysteine and methionine are present in relatively lower quantities [55,56,57]. Furthermore, *Auricularia auricula* is notable for its substantial mineral content, which includes essential elements such as calcium (Ca), potassium (K), magnesium (Mg), sodium (Na), phosphorus (P), and iron (Fe), with iron levels potentially reaching as high as 285 μg/g [55,58,59]. In terms of vitamins, this fungus is a rich source of several important nutrients, including Vitamin A, Vitamin D_2_, and Vitamin K [60].

### 7.2. Pharmacological Effects

*Auricularia auricula*, commonly known as the wood ear mushroom, has a rich history of medicinal application that is well-documented in historical texts such as the Xinglun Yaoun (*Treatise on Medicinal Properties*) and the *Tang Materia Medica*. These texts delineate the mushroom’s associations with specific tree species and its purported therapeutic effects, which include the dispelling of wind, alleviation of blood stasis, and treatment of hemorrhoids. Recent scientific investigations have elucidated some of the bioactive properties of *Auricularia auricula-judae*. For instance, studies have demonstrated that dichloromethane extracts of this mushroom can inhibit the production of nitric oxide (NO) and reduce the secretion of inflammatory cytokines, including interleukin-6 (IL-6), tumor necrosis factor-alpha (TNF-α), and IL-1β, in lipopolysaccharide (LPS)-stimulated macrophages [61]. Furthermore, Linoleic acid and oleic acid, extracted from *Auricularia auricula-judae* using ethanol, have been shown to bind directly to the kinase domain of Tropomyosin receptor kinase B (TrkB), thereby effectively blocking downstream signaling pathways triggered by BDNF (the ligand for TrkB), including the Akt and MAPK pathways. This results in reduced proliferation and increased apoptosis (programmed cell death) in cancer cells [62]. Moreover, *Auricularia auricula* is rich in various bioactive constituents, such as polysaccharides, melanin, proteins, and polyphenolic compounds, all of which are associated with a wide array of pharmacological activities. The exploration of these compounds holds promise for advancing our understanding of their therapeutic potential and applications in contemporary medicine.

*Auricularia auricula* polysaccharide (AAP) exhibits notable hypoglycemic effects through the modulation of critical enzymes involved in glucose metabolism. Specifically, AAP facilitates the phosphorylation of glycogen synthase kinase 3β (GSK3β), thereby promoting hepatic glycogen synthesis. Concurrently, it inhibits the activity of phosphoenolpyruvate carboxykinase (PEPCK) and glucose-6-phosphatase (G6Pase), which enhances the phosphorylation of forkhead box O1 (FOXO1). This process effectively suppresses hepatic gluconeogenesis, leading to a reduction in blood glucose levels [63]. Additionally, the β-(1 → 3)-D-glucan backbone of AAP contributes to its antitumor activity [64]. The abundant presence of β-(1 → 3)-D-glucans in AAP has also led to the development of AAP-based nanoparticles as carriers for antitumor drug delivery [65].

In vitro studies have demonstrated that AAP-chitosan nanoparticles (AAP-CS-NPs) exhibit significant cytotoxic effects against MCF7 breast cancer cells [66]. Research conducted in 2017, ref. [67] revealed that oral administration of AAP at doses of 100 or 400 mg/kg over a four-week period significantly reduced blood glucose levels by enhancing glucose metabolism. The study further elucidated that the hypoglycemic effects of AAP are, at least in part, mediated by the regulation of the antioxidant system and the activation of the NF-κB signaling pathway. Furthermore, investigations conducted in 2017, ref. [68] demonstrated that hydrolysates of AAP (APSHs) significantly elevated both hepatic glycogen and pancreatic insulin levels, indicating substantial antidiabetic properties.

AAP extracted from *Auricularia auricula* has also been shown to effectively inhibit nutritional obesity by modulating gut microbiota and metabolic pathways. It appears to improve the gut microbial environment, potentially by influencing the growth of bacteria that produce short-chain fatty acids (SCFAs), thereby mitigating liver injury and addressing hyperlipidemia. Other fungal dietary fibers have been observed to upregulate SCFA production, leading to pronounced hypolipidemic effects [69]. Acidic forms of AAP have exhibited properties that reduce the expression of inflammatory factors and alleviate liver histomorphological changes induced by a high-fat, high-fructose diet (HFFD). Subsequent studies indicate that AAP downregulates hepatic pro-lipidogenic genes while upregulating genes associated with lipolytic and mitochondrial activity, thereby improving lipid metabolism and reducing the risks of metabolic disorders and obesity [70].

Furthermore, AAP and its derivatives have been shown to interact directly with specific pattern recognition receptors, such as TLR4 and Dectin-1, and activate immune-related signaling pathways, including the NF-κB and Syk-dependent pathways. These interactions promote the transcription of downstream target genes and enhance the release of immune-related cytokines, as well as the maturation of T and B lymphocytes [71]. A study conducted in 2015 by Cai, Lin, Luo, Liang, and Sun established the antimicrobial potential of AAP against Escherichia coli and Staphylococcus aureus [72]. Their findings suggested that AAP can attenuate the secretion of epidermal growth factor receptor (EGFR) ligands (such as TGF-α, EGF, and AREG) and modulate signaling pathways between the gut and lungs, effectively inhibiting the EGFR/JNK signaling pathway in lung tissues, thereby conferring therapeutic effects against silicosis [73].

The anticoagulant activity of AAP has also been investigated, demonstrating that it involves thrombin inhibition mediated by antithrombin. AAP primarily exhibits its anticoagulant effects by inhibiting platelet aggregation and prolonging blood coagulation [36]. Additionally, ref. [58] discovered that pulsed electric field-assisted extraction at a field strength of 24 kV/cm can significantly enhance the anticoagulant properties of AAP [74]. Research by [75] revealed that acid hydrolysates of AAP (AAPs-F) prolonged the lifespan and activated the antioxidant system of *Caenorhabditis elegans*, suggesting notable antioxidant capacity in vivo. Interestingly, ultrasonically modified and degraded AAP has shown enhanced bioactive properties, significantly improving antioxidant, hypoglycemic, and hypolipidemic activities [76]. As indicated by [77], solution plasma processing (SPP) and H_2_O_2_ degradation significantly improved AAP’s metal-chelating and DPPH radical-scavenging activities. Moreover, SPP irradiation resulted in a decrease in the characteristic viscosity and an enhancement of AAP’s biological activity. In 2019, ref. [78] isolated a polysaccharide known as SNAAP from the ascospores of *Auricularia auricula* using hot water extraction. This polysaccharide primarily comprises glucose and mannose in a 1:1 ratio. The administration of SNAAP attenuated radiation-induced disorders in glucose metabolism and regulated glucose homeostasis in Kunming mice. This regulatory effect is mediated through the activation of the Akt/GSK-3β/GYS2 axis, inhibition of the JNK/Akt/FOXO1 axis, and stimulation of the PDX1/GLUT2/IRS1 axis to promote insulin secretion.

In addition to its diverse biological activities, AAP also shows promise as a valuable ingredient in whey protein beverages, providing pseudoplasticity, favorable zeta potential, well-ordered microstructures, and distinct sensory characteristics [79]. While the broad therapeutic potential of AAP is clear, further investigation into the specific mechanisms of action of its active components is warranted. Continued research in this area holds great promise for unlocking additional applications and benefits associated with AAP.

*Auricularia auricula* melanin (AAM) exhibits remarkable antioxidant properties, effectively scavenging various free radicals, including ABTS^+^, DPPH, and hydroxyl (OH) radicals. Moreover, it significantly mitigates cellular oxidative damage induced by hydrogen peroxide (H_2_O_2_) [80]. A study conducted by [81] demonstrated that melanin derived from the fermentation of *Auricularia auricula* possesses notable Fe^2+^ chelating capacity, along with DPPH and superoxide radical scavenging activities, underscoring its strong antioxidant potential.

In terms of hepatoprotection, AAM has been shown to significantly decrease alcohol-induced elevations in serum triglycerides (TG), total cholesterol (TC), gamma-glutamyl transpeptidase (GGT), hepatic malondialdehyde (MDA), low-density lipoprotein cholesterol (LDL-C), aspartate aminotransferase (AST), and alanine aminotransferase (ALT) in murine models. It also effectively curtails hepatic lipid accumulation and steatosis while enhancing the activity of crucial enzymes such as alcohol dehydrogenase (ADH), catalase (CAT), and superoxide dismutase (SOD) [82,83]. Insights from metabolomics have revealed that AAM mitigates alcohol-induced liver injury by modulating hepatic metabolic pathways, including phosphatidylinositol signaling, starch/sucrose metabolism, glycolysis, and the biosynthesis of unsaturated fatty acids [82,83].

Furthermore, AAM significantly enhances the composition of gut microbiota in alcohol-exposed mice, particularly by increasing the abundance of beneficial bacterial genera such as *Akkermansia* and *Bifidobacterium* [83]. Research conducted by [84] revealed that melanin from *A. auricula* substantially inhibits biofilm formation in various bacterial strains, including *Escherichia coli* K-12, *Pseudomonas aeruginosa* PAO1, and *Pseudomonas fluorescens* P-3. Additionally, arginine-modified AAM demonstrated improved antibacterial activity against *Staphylococcus aureus*, likely through the inhibition of bacterial proliferation and disruption of cellular structures, thus highlighting its potential as a candidate for antibacterial agent development [85].

Despite the multifaceted bioactivities exhibited by AAM—encompassing antioxidant, hepatoprotective, and antimicrobial properties—several critical translational barriers remain. Firstly, the specific chromophores and polymeric configurations responsible for its radical scavenging capacity remain poorly characterized, impeding potential optimization of its activities. Secondly, although metabolic pathway regulation has been observed, the causal relationships linking AAM to specific molecular targets remain unestablished, with the epigenetic and immunomodulatory effects yet to be explored. Lastly, the suboptimal solubility and bioavailability of unmodified AAM hinder its therapeutic efficacy, while the antibacterial mechanisms of arginine-modified AAM lack structural validation. Future research endeavors must employ combinatorial chemistry to elucidate structure-activity relationships, integrate multi-omics approaches to characterize AAM-host-microbiome interactions, and develop innovative nano-encapsulation strategies to facilitate clinical translation.

The crude protein content of dried *A. auricula* (ranging from 6.5% to 13%) is lower than that found in many other wild and cultivated mushroom species [86,87]. Nonetheless, the proteins derived from this fungus exhibit substantial medicinal and pharmaceutical properties [88,89]. Notably, immunomodulatory proteins, such as APP, play a crucial role in activating macrophages, which in turn induce the production of nitric oxide (NO) and tumor necrosis factor-alpha (TNF-α), while also demonstrating hemagglutination activity [88]. Lectins, which are also present in *A. auricula*, contribute to the maintenance of respiratory flora balance, enhance biological barriers, and mitigate the colonization of pathogenic bacteria. Moreover, these molecules have shown potential in anti-tumor applications [90].

The glycoprotein AAG-3, specifically isolated from *A. auricula*, demonstrates hypoglycemic potential by facilitating cellular glycogen synthesis and inhibiting gluconeogenesis in insulin-resistant cells, suggesting its potential as a functional supplement for individuals with diabetes [91]. Importantly, both lectins and immunoproteins, such as ABL and APP, exhibit remarkable stability characteristics, including resistance to heat and freeze conditions, as well as tolerance to varying acid and alkaline environments. These properties, along with their dehydration stability, position them as promising candidates for use as immunostimulants in therapeutic applications [90].

Research conducted by [92] has demonstrated that ethanolic extracts of *A. auricula* significantly reduce total cholesterol levels and the atherosclerotic index in murine models, an effect closely linked to its rich polyphenolic composition. *A. auricula* is well-known for its abundance of phenolic and flavonoid compounds, which possess notable antioxidant properties through mechanisms such as free radical scavenging and the inhibition of oxidative stress [93,94]. Notably, the concentration of polyphenolic compounds extracted from *A. auricula* utilizing the ACAP method surpasses that of other mushroom varieties, yielding superior antioxidant activity [95]. These findings underscore the considerable potential of *A. auricula* as both a pharmaceutical and dietary supplement, attributed to its hypolipidemic and antioxidant characteristics.

The pharmacological effects and mechanisms of action of the active components in *A. auricula* are summarized in Table 2 and Figure 4.

## 8. Industrialized Applications of *Auricularia* Species

The escalating consumer demand for safe and effective natural products has catalyzed the burgeoning utilization of *A. auricula* extracts within the pharmaceutical and cosmetic sectors [96]. Rich in polyphenolic compounds, *A. auricula* serves as a prominent ingredient in natural skincare formulations aimed at combating premature skin aging, attributed to its potent antioxidant and anti-aging properties [97].

In the realm of drug delivery research, the complexation of *A. auricula* polysaccharides (AAP) with casein, induced by glucono-δ-lactone, yields composite gels that act as effective carriers for curcumin. This innovative system facilitates controlled release and enhances bioavailability [98]. Building on this foundational work, ref. [99] have developed histidine-modified AAP micelles (His-AAP-PTX) to achieve efficient loading of paclitaxel (PTX). Both in vitro and in vivo studies have demonstrated a markedly superior antitumor efficacy when compared to free PTX. Furthermore, polyelectrolyte complex nanoparticles (PEC-NPs), formulated through electrostatic interactions between AAP and chitosan, exhibit a high loading capacity and cellular uptake efficiency for doxorubicin hydrochloride (DOX-HCl), simultaneously mitigating toxicity to normal cells [66].

Beyond its applications in drug delivery, *A. auricula* displays significant promise in the development of advanced materials. For instance, nitrogen–phosphorus double-doped porous spore carbon (NP-PSC) derived from *A. auricula* has emerged as an efficient electrode material for lithium-sulfur batteries, optimizing electrochemical performance beyond that of conventional activated carbon electrodes [100]. Additionally, *A. auricula* melanin (AAM), harnessing its UV-absorbing properties, demonstrates potential as an eco-friendly colorant for the glass and plastic industries, thereby enhancing UV resistance in materials [101]. Comparative studies have indicated that films derived from AAP possess superior thermal stability and enhanced antibacterial properties over chitosan films, effectively extending the shelf life of fresh beef by approximately four days [102]. Notably, *A. auricula* also exhibits significant chelating effects on lead ions, with in vivo investigations indicating its detoxification potential in renal and splenic tissues, which positions it as a candidate for heavy metal detoxification [103]. Applications of *A. auricula* in industry are summarized in Figure 5.

Despite these advancements, the specific structures and mechanisms of action of many active components within *A. auricula* remain to be fully elucidated. Thus, multidisciplinary research endeavors are essential to comprehensively explore its potential in the synthesis of novel functional materials, innovative pharmaceuticals, and functional foods.

## 9. Conclusions 

The genus *Auricularia*, particularly *A. auricula*, represents a unique convergence of nutritional, medicinal, and industrial value. This review synthesizes its taxonomic complexity, bioactive compounds (polysaccharides, melanin, proteins, and polyphenols), and multifaceted applications, revealing a species whose potential is both vast and underexploited. Despite advancements in molecular taxonomy and cultivation, critical gaps persist—such as mechanistic insights into bioactive compound functions, sustainable substrate alternatives, and the ecological impact of spent substrate disposal. The pharmacological properties of *A. auricula* compounds, including hypoglycemic, antitumor, and immunomodulatory effects, are well-documented yet lack translational depth, particularly in clinical and industrial contexts. Furthermore, while genomic studies have resolved phylogenetic ambiguities, functional annotation of key genes (e.g., laccases, glycoside hydrolases) remains superficial, limiting strain engineering for bioremediation or nutraceutical enhancement.

Future research must adopt a systems-level approach to unlock *Auricularia*’s full potential. First, integrating multi-omics (genomics, transcriptomics, proteomics) will elucidate the genetic and metabolic networks governing bioactive compound synthesis, enabling targeted manipulation for yield and efficacy optimization. Second, sustainable cultivation demands innovation: AI-driven substrate formulation, synthetic microbial consortia for lignocellulosic waste upcycling, and gene-edited strains resilient to environmental stressors. Third, translational pipelines should bridge lab-scale discoveries to industrial applications—for instance, melanin-based nanomaterials for antimicrobial coatings or polysaccharide-drug conjugates for targeted cancer therapy. Climate-smart cultivation, leveraging geographic distribution data to predict adaptive traits, will be critical as global temperatures rise. Lastly, interdisciplinary collaboration is essential to address socioeconomic barriers, such as cost-effective SMS recycling and policy frameworks for circular bioeconomy integration. By prioritizing these avenues, *A. auricula* could emerge as a model organism for sustainable biotechnology, aligning food security with ecological resilience.

This review underscores *Auricularia* species as resource of profound significance, bridging traditional knowledge with modern scientific inquiry. From its taxonomic redefinition to its pharmacological versatility and industrial adaptability, the species exemplifies the untapped potential of fungal biodiversity. Yet, its journey from germplasm to global impact hinges on resolving mechanistic mysteries, adopting sustainable practices, and fostering cross-sector innovation. As research transcends descriptive biology to embrace functional and applied science, *A. auricula* stands poised to redefine its role—not merely as a functional food but as a cornerstone of next-generation biotechnological solutions. The path forward demands rigor, creativity, and collaboration to transform this ancient organism into a beacon of sustainable progress.

## Figures and Tables

**Figure 1 biology-14-01555-f001:**
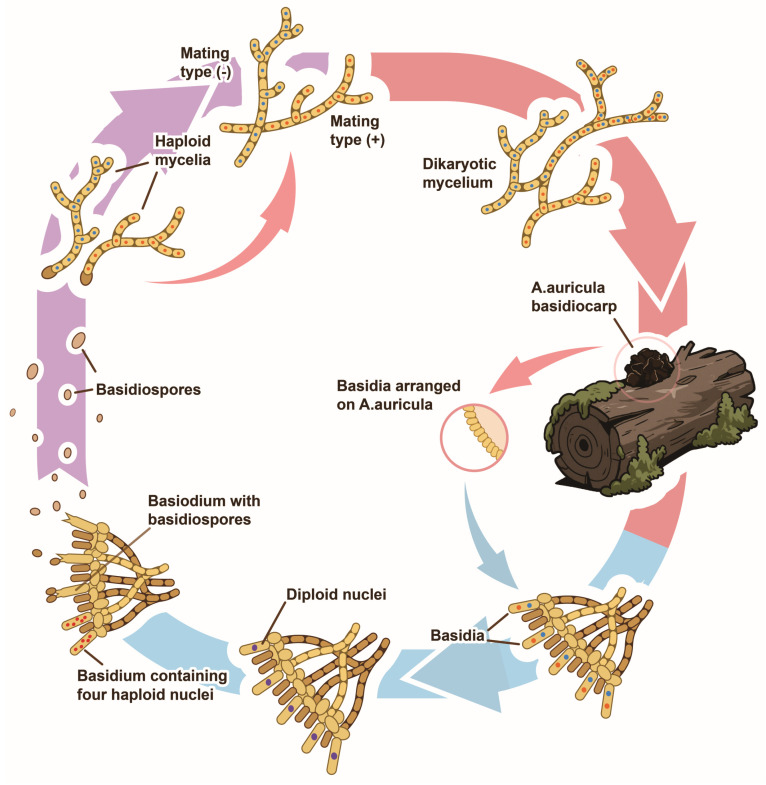
The lifecycle of *Auricularia* species.

**Figure 2 biology-14-01555-f002:**
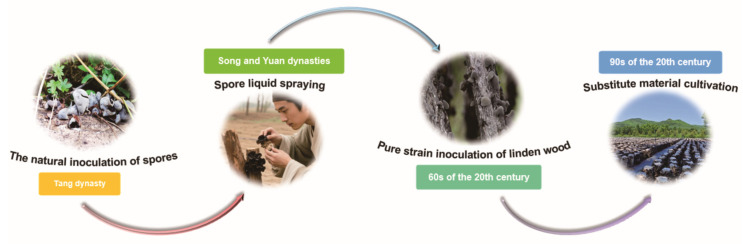
The history of cultivation and domestication of *Auricularia* species.

**Figure 3 biology-14-01555-f003:**
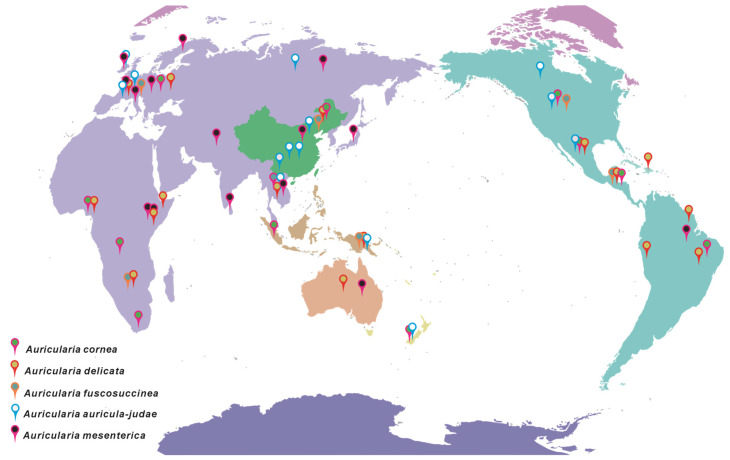
Global distribution of *Auricularia* species complexes.

**Figure 4 biology-14-01555-f004:**
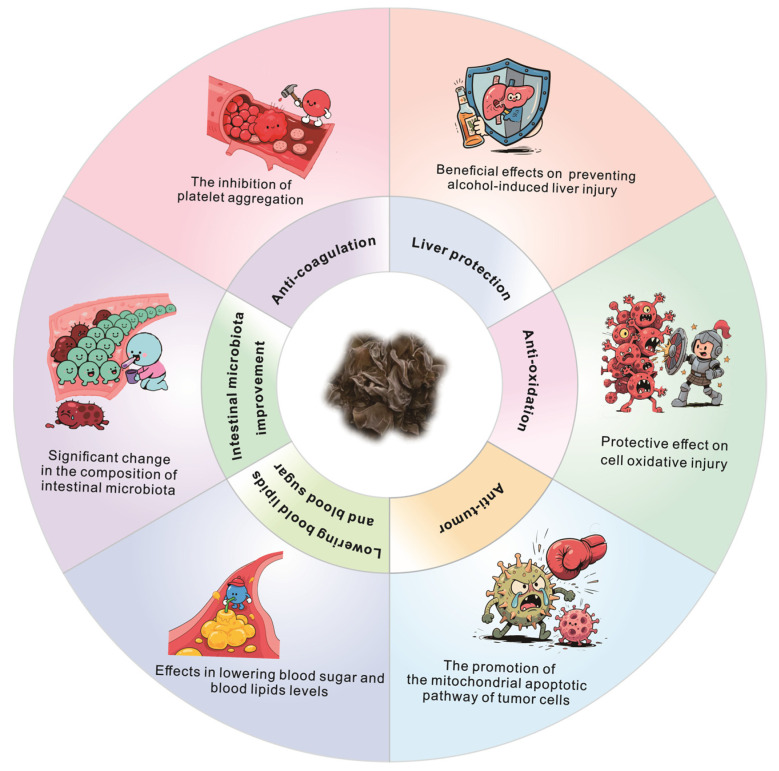
The biological activity and medicinal efficacy of *Auricularia* species.

**Figure 5 biology-14-01555-f005:**
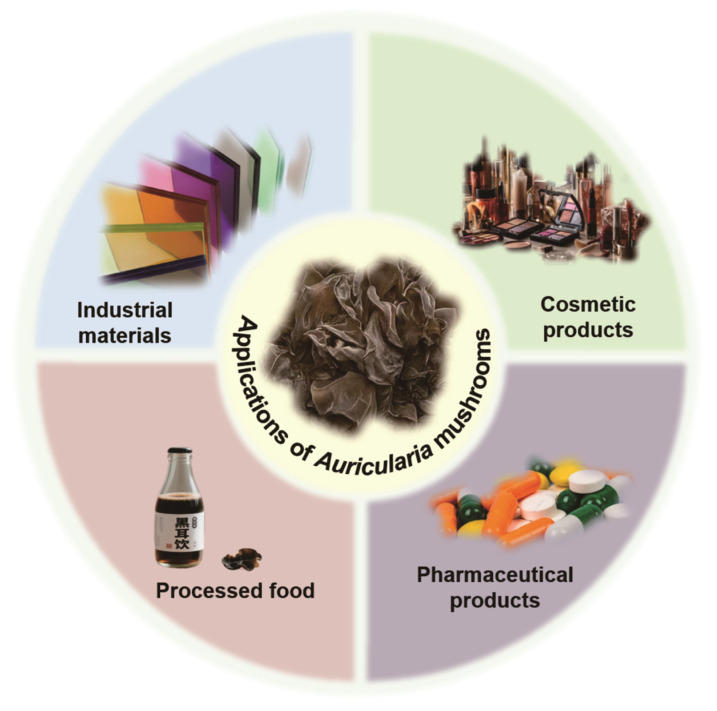
The industry application of *Auricularia* species.

**Table 1 biology-14-01555-t001:** Geographic Distribution of Major *Auricularia* Species Complexes.

Complex Name	Species Name	Country
*A. auricula-judae*	*A. tibetica*	China
	*A. americana*	China, Russia, USA
	*A. angiospermarum*	USA
	*A. heimuer*	China, Japan, Russia
	*A. auricula-judae*	France, Czech Republic, UK, Germany
	*A. villosula*	China, Russia, Thailand
	*A. minutissima*	China
*A. cornea*	*A. cornea*	China, Brazil, Singapore, Vietnam, Sri Lanka
	*A. novozealandica*	New Zealand
*A. delicata*	*A. delicata*	Cameroon
	*A. sinodelicata*	China
	*A. australiana*	Australia
	*A. lateralis*	China
	*A. conferta*	Australia
*A. fuscosuccinea*	*A. fibrillifera*	China, Papua New Guinea, Zambia
	*A. thailandica*	China, Thailand
	*A. fuscosuccinea*	USA, Brazil
	*A. pilosa*	Ethiopia, Tanzania, Australia, Zambia
	*A. scissa*	Dominican Republic
	*A. subglabra*	Brazil, Costa Rica, French Guiana
	*A. nigricans*	Costa Rica, USA, Mexico
	*A. camposii*	Brazil
	*A. tremellosa*	Mexico, Peru, Brazil
*A. mesenterica*	*A. brasiliana*	Brazil
	*A. asiatica*	China, Indonesia
	*A. mesenterica*	France, Czech Republic, Estonia, Switzerland, UK, Italy, Uzbekistan
	*A. africana*	Kenya, Uganda
	*A. orientalis*	China
	*A. srilankensis*	Sri Lanka

**Table 2 biology-14-01555-t002:** Bioactive components of *Auricularia* species and their functions.

Compound Type	Representative Component	Bioactivity	Mechanism/Key Effects	Study Model	References
Polysaccharides	AAP	Hypoglycemic	Activates GSK3β phosphorylation → hepatic glycogen synthesis; inhibits PEPCK/G6Pase → suppresses gluconeogenesis	Mouse models	[63,67,68]
		Antitumor	β-(1 → 3)-D-glucan backbone mediates direct antitumor effects; nanoparticle carriers enhance drug delivery	MCF7 cells	[64,65,66]
		Gut microbiota modulation	Promotes SCFA-producing bacteria → improves lipid metabolism	Mouse models	[69,70]
		Anti-obesity	Downregulates hepatic lipogenic genes; upregulates lipolytic/mitochondrial activity genes	HFFD-fed mice	[70]
		Immunomodulation	Binds TLR4/Dectin-1 → activates NF-κB/Syk pathways → promotes lymphocyte maturation	Immune cells in vitro	[71]
		Antimicrobial	Inhibits *E. coli*/*S. aureus*; modulates gut-lung EGFR/JNK axis → anti-silicosis	Cells/mouse models	[72,73]
		Anticoagulant	Inhibits thrombin activity; delays platelet aggregation	Blood models in vitro	[36,58,74]
		Antioxidant	Extends lifespan of *C. hidradii* nematodes; activates antioxidant systems	Nematode model	[75]
	SNAAP	Radioprotective	Regulates Akt/GSK-3/GYS2 (glycogen synthesis), JNK/Akt/FOXO1 (gluconeogenesis), PDX1/GLUT2/IRS1 (insulin secretion) axes	Irradiated mice	[78]
Melanins	AAM	Antioxidant	Scavenges ABTS^+^/DPPH/OH radicals; repairs H_2_O_2_-induced cellular damage	Cell models	[80,81]
		Alcohol-induced liver protection	Reduces serum TG/TC/ALT/AST; enhances the activity of ADH/CAT/SOD; regulates the PI signaling pathway and glucose metabolism.	Alcohol-injured mice	[82,83]
		Gut microbiota modulation	Increases probiotics such as *Akkermansia* and *Bifidobacterium*	Alcohol-exposed mice	[83]
	Arginine-modified AAM	Antibacterial	Disrupts biofilms/cellular structures of *S. aureus*	Bacterial strains in vitro	[84,85]
Proteins	APP	Immunomodulation	Activates macrophages to release NO/TNF-α; blood coagulation activity	RAW264.7 cells	[88]
	Lectins (e.g., ABL)	Antitumor/Antibacterial	Balances respiratory flora; inhibits pathogenic bacterial colonization	In vitro models	[90]
	Glycoprotein AAG-3	Hypoglycemic	Promotes glycogen synthesis in insulin-resistant cells; inhibits gluconeogenesis	Insulin-resistant cells	[91]
Polyphenols	Phenolics/Flavonoids	Hypolipidemic	Reduces total cholesterol and atherosclerotic index	Hyperlipidemic mice	[92]
		Antioxidant	Free radical scavenging; oxidative stress inhibition	In vitro chemical assays	[93,94,95]

## Data Availability

Data sharing is not applicable. No new data were created or analyzed in this study.

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
