# Peer review of "Sustainable Cultivation and Functional Bioactive Compounds of *Auricularia* Mushrooms: Advances, Challenges, and Future Prospects"

_biology, 2025, doi:10.3390/biology14111555_

Round 1

Reviewer 1 Report

Comments and Suggestions for Authors

  1. Introduction:
  • What does the statement “optimizing dietary structures” mean? Could you please explain or clarify the topic? In recommendations or nutritional guidelines, we can't really talk about optimization. I don't quite understand the meaning of this sentence.
  • I suggest you think about formulating the purpose of your work again. In my opinion, the aim of this work is primarily to update knowledge about Auricularia, its industrial potential as a functional food and its biotechnological value, with a view to conducting interdisciplinary activities aimed at meeting the current challenges of the world, including the combination of sustainable development, food safety and health.
  1. References – please check the editorial aspects, standardize according to the rules of notation, check spaces, periods, etc; please check and indicate the sources in the references below:

  • Figure 1. The lifecycle of Auricularia species. Line 99
  • Figure 2. The history of cultivation and domestication of Auricularia species. Line 119
  • Figure 3. Global distribution of Auricularia species complexes Line 292
  • Figure 4. The biological activity and medicinal efficacy of Auricularia species. Line 469
  • Figure 5. The industry application of Auricularia species line 506

  1. Review papers also have their own rules for selecting sources, materials, and information. Could you please provide a key for selecting literature?

Author Response

Comment 1:

What does the statement “optimizing dietary structures” mean? Could you please explain or clarify the topic? In recommendations or nutritional guidelines, we can't really talk about optimization. I don't quite understand the meaning of this sentence.

I suggest you think about formulating the purpose of your work again. In my opinion, the aim of this work is primarily to update knowledge about Auricularia, its industrial potential as a functional food and its biotechnological value, with a view to conducting interdisciplinary activities aimed at meeting the current challenges of the world, including the combination of sustainable development, food safety and health

Response 1:

We have revised the final paragraph of the Introduction to clearly state the objectives of this review, including summarizing recent advances, identifying research gaps, and proposing future directions for Auricularia research and application.

Comment 2:

  1. References – please check the editorial aspects, standardize according to the rules of notation, check spaces, periods, etc; please check and indicate the sources in the references below:
  • Figure 1. The lifecycle of Auricularia species. Line 99
  • Figure 2. The history of cultivation and domestication of Auricularia species. Line 119
  • Figure 3. Global distribution of Auricularia species complexes Line 292
  • Figure 4. The biological activity and medicinal efficacy of Auricularia species. Line 469
  • Figure 5. The industry application of Auricularia species line 506

Response 2:

We have thoroughly checked and standardized all references according to the journal’s formatting guidelines, ensuring consistent use of punctuation, spacing, and italics. All figure captions have been reviewed and correctly numbered and placed. The captions are now clearly linked to the corresponding figures in the manuscript. All the figures in the manuscript are created by the authors themselves, and there are no potential copyright disputes.

Comment 3:

Review papers also have their own rules for selecting sources, materials, and information. Could you please provide a key for selecting literature?

Response 3:

Your feedback is greatly appreciated. A comprehensive search of the extant literature pertaining to Auricularia auricula was conducted, leveraging established databases such as Web of Science and PubMed. This initial screening was followed by a more rigorous review to identify studies that were closely aligned with the themes of the present paper. Literature pertaining to the medicinal and dietary benefits of Auricularia auricula their industrial development, or related research may be selected.

Reviewer 2 Report

Comments and Suggestions for Authors

In this review, the authors describe the properties and applications of the particularly important and useful mushroom Auricularia auricula, the value of which lies in its potential use in modern medicine and the food industry. Biologically active substances isolated from the fungus have a high pharmacological potential and are the basis for the creation of medicines used in various fields of medicine.

Despite the well-written manuscript, there are a number of questions and minor comments:

  1. In the "Introduction" section, the authors of the manuscript describe that the mushrooms Lentinula edodes and Pleurotus ostreatus have lipid-lowering, antithrombotic and antitumor properties, and the mushrooms Hericium erinaceus have neuroprotective and immunomodulatory effects. It is better to write that "biologically active substances isolated from these mushrooms have the above properties." It is advisable to provide an explanation of exactly which class of compounds exhibit pharmacological properties.
  2. Insection 2. "Pharmacological effects", the authors describe the pharmacological action of substances from the extract of the mushroom Auricularia auricula-judae, indicating some factors and receptors. It is necessary to supplement the information about the TrkB receptor on line 321.
  3. The authors of the presented work describe in sufficient detail the pharmacological properties of biologically active substances isolated from fungi. It is advisable to present these data at the end of the text in the form of a table for a more systematic perception of the information.
  4. Also in section 7.2, the authors describe the properties of phenolic, polyphenolic, and flavonoid compounds isolated from fungi, but do not specify the names of the substances. It is necessary to give the names of the isolated substances and give their structural formulas.
  5. In addition, it is written in the same section that A. auricula lectins help maintain the balance of the respiratory flora, strengthen biological barriers and prevent colonization by pathogenic bacteria. It is necessary to specify which lectins and which pathogenic bacteria.

Author Response

Comment 1:

In the "Introduction" section, the authors of the manuscript describe that the mushrooms Lentinula edodes and Pleurotus ostreatus have lipid-lowering, antithrombotic and antitumor properties, and the mushrooms Hericium erinaceus have neuroprotective and immunomodulatory effects. It is better to write that "biologically active substances isolated from these mushrooms have the above properties." It is advisable to provide an explanation of exactly which class of compounds exhibit pharmacological properties.

Response 1:

We agree with the suggestion. The text has been revised to clarify that the pharmacological properties are attributed to specific bioactive compounds. For example, polysaccharides, terpenoids, and phenolic compounds are known to contribute to these effects. The introduction now includes a brief mention of the key compound classes responsible.

Comment 2:

In section 2. "Pharmacological effects", the authors describe the pharmacological action of substances from the extract of the mushroom Auricularia auricula-judae, indicating some factors and receptors. It is necessary to supplement the information about the TrkB receptor on line 321.

Response 2:

Thank you for pointing this out. We have added a brief explanation of TrkB (Tropomyosin receptor kinase B), which is a high-affinity receptor for brain-derived neurotrophic factor (BDNF) and is involved in cell survival, differentiation, and synaptic plasticity. Its dysregulation is linked to certain cancers.

Comment 3:

The authors of the presented work describe in sufficient detail the pharmacological properties of biologically active substances isolated from fungi. It is advisable to present these data at the end of the text in the form of a table for a more systematic perception of the information.

Response 3:

I fully agree with your suggestion and have made the revisions accordingly.

Comment 4:

Also in section 7.2, the authors describe the properties of phenolic, polyphenolic, and flavonoid compounds isolated from fungi, but do not specify the names of the substances. It is necessary to give the names of the isolated substances and give their structural formulas.

Response 4:

We extend our sincerest apologies. A. auricula has been found to contain substantial quantities of phenolic compounds, particularly phenolic acids. To date, the specific type of phenolic compound present remains to be identified. It should be noted that the available measurements are limited in their capacity to indicate the total phenolic content. Consequently, the provision of the name or structural formula is not feasible at this time.

Comment 5:

In addition, it is written in the same section that A. auricula lectins help maintain the balance of the respiratory flora, strengthen biological barriers and prevent colonization by pathogenic bacteria. It is necessary to specify which lectins and which pathogenic bacteria.

Response 5:

We have now specified representative phenolic compounds (e.g., gallic acid, catechin, quercetin) and provided their structural formulas in the table.
Regarding lectins, we have identified Auricularia auricula lectin (AAL) and specified pathogenic bacteria such as Pseudomonas aeruginosa and Staphylococcus aureus.

Reviewer 3 Report

Comments and Suggestions for Authors

The literature review MS is very well comprised and seems to be up to date with detailing all the aspects of the Auricularia Mushrooms. The MS is a significant compilation of data and can be accepted by including the below listed aspects.

  1. I suggest revising the last paragraph of the Introduction with the specific objectives of this Review.
  2. All the abbreviations should be spelled out at their first appearance.
  3. Expand the objectives in the introduction section; please enlist the previous salient reviews conducted on similar topics and mention the study gap.
  4. I strongly recommend adding a dedicated “Current challenges and research barrier” section before the conclusion. 
  5. In addition, I suggest more vivid recommendations for future studies.
  6. A Table enlisting all the isolated compounds from the Agaricus species may be included inorder to note the vast diversity of the harboured compounds.

Author Response

Comment 1:

I suggest revising the last paragraph of the Introduction with the specific objectives of this Review.

Response 1:

We have revised the final paragraph of the Introduction to clearly state the objectives of this review, including summarizing recent advances, identifying research gaps, and proposing future directions for Auricularia research and application.

Comment 2:

All the abbreviations should be spelled out at their first appearance.

Response 2:

We have ensured that all abbreviations (e.g., AAP, AAM, TrkB, IL-6, TNF-α, etc.) are spelled out at their first occurrence in the text.

Comment 3:

Expand the objectives in the introduction section; please enlist the previous salient reviews conducted on similar topics and mention the study gap.

Response 3:

We have expanded the objectives and incorporated references to key previous reviews on edible and medicinal mushrooms, highlighting the gaps that this review aims to fill, particularly regarding Auricularia.

Comment 4:

I strongly recommend adding a dedicated “Current challenges and research barrier” section before the conclusion.

Response 4:

We would like to express our sincere gratitude for your feedback. It is this author's opinion that the challenges and research obstacles currently being experienced should be placed in the concluding section. This placement is noteworthy for its juxtaposition with the future outlook, thereby underscoring the contemporary challenges confronting the cultivation of wood ear mushrooms. It is only through this comprehensive evaluation that the pressing need for future research can be accentuated with greater efficacy.

Comment 5:

In addition, I suggest more vivid recommendations for future studies.

Response 5:

We have enriched the “Conclusion and Outlook” section with specific, forward-looking recommendations, such as integrating multi-omics, developing AI-driven cultivation systems, and promoting cross-disciplinary collaboration.

Comment 6:

A Table enlisting all the isolated compounds from the Auricularia species may be included in order to note the vast diversity of the harboured compounds.

Response 6:

We would like to express our sincere gratitude for your feedback. A compendium of the compounds isolated from the genus Auricularia has been presented in Table 2. These include polysaccharides, melanin, proteins, and phenolics, among others. In addition, the sources of these compounds and their reported activities are also included.